# Automatic muscle segmentation on healthy abdominal MRI using nnUNet

**Victoria Joppin**[*1,2]                          VICTORIA.JOPPIN@UNIV-EIFFEL.FR
**Niamh Belton**[3,4]                              NIAMH.BELTON@UCDCONNECT.IE
**Marc-Adrien Hostin**[5]                          MARC-ADRIEN.HOSTIN@UNIV-AMU.FR
**Marc-Emmanuel Bellemare**[5]          MARC-EMMANUEL.BELLEMARE@UNIV-AMU.FR
**Aonghus Lawlor**[6]                              AONGHUS.LAWLOR@UCD.IE
**Kathleen Curran**[3]                             KATHLEEN.CURRAN@UCD.IE
**Thierry Bege**[1,7]                              THIERRY.BEGE@AP-HM.FR
**Catherine Masson**[1]                     CATHERINE.MASSON@UNIV-EIFFEL.FR
**David Bendahan**[2]                          DAVID.BENDAHAN@UNIV-AMU.FR

[1] *Aix Marseille Université, Université Gustave Eiffel, LBA, Marseille France*

[2] *Aix Marseille Université, CNRS, CRMBM UMR 7339, Marseille France*

[3] *School of Medicine, University College Dublin, Dublin, Ireland*

[4] *Science Foundation Ireland Centre for Research Training in Machine Learning*

[5] *Aix Marseille Université, Université de Toulon, CNRS, LIS UMR 7020, Marseille France*

[6] *School of Computer Science, University College Dublin, Dublin, Ireland*

[7] *Aix Marseille Université, Département de Chirurgie, Hôpital Nord, APHM, Marseille, France*

**Editors:** Accepted for publication at MIDL 2024

## Abstract

Understanding the dynamics of the abdominal wall is essential in both physiology and surgery. To study the mechanical functionality of the abdominal wall, segmentation of the abdominal muscles could be useful but is a manual, tedious and time-consuming process. In this study, we assessed the capability of Deep Learning to automatically segment the abdominal muscles from the axial plane of a unique dynamic abdominal MRI (2D+t) dataset. The 2D slices were acquired while the subject performed various exercises. The State-of-the-Art segmentation model, nnUNet was trained on $5,492$ images from fifteen healthy subjects and tested it on $1,801$ images from five different subjects. We assessed the segmentation accuracy using DICE similarity coefficient, Hausdorff distance, as well as motion of the abdominal muscles. The ground truth and nnUNet segmentation showed high concordance, with a DICE over 0.87 for all exercises and muscles, and minimal differences in abdominal muscles motion. nnUNet effectively automates abdominal muscle segmentation, offering efficiency and new clinical applications in abdominal physiology.

**Keywords:** Medical imaging segmentation, Deep learning, Abdominal wall muscles

## 1. Introduction

Abdominal wall pathologies present a clinical challenge due to the high prevalence of hernia recurrence, which can reach up to 30% after repair (Romain *et al.*, 2020). Hernias are a frequent medical condition that occurs when abdominal contents protrude through a defect in the abdominal wall. Imaging techniques with precise segmentation of structures are

---

* Coorresponding author

often necessary to understand and treat this condition, as they allow for in-depth analysis of the anatomy, physiological and pathological response of the abdominal wall. At present, researchers manually segment each abdominal muscle to quantify their mechanical functionality. Given that over 300 slices is acquired per subject, this is a tedious and costly process. To the best of our knowledge, this is the first study to explore the automatic segmentation of abdominal muscles from MRI images. We leverage the State of the Art nnUNet due to its superior performance for biomedical image segmentation (Isensee *et al.*, 2021).

## 2. Methods

### 2.1. Dataset

The dataset is composed of 60 dynamic MRI volumes (2D+T), consisting of a 2D slice of the abdominal region (Figure 1-a) acquired on a 3T MRI scanner in the supine position at L3-L4 disc level over time. Twenty healthy subjects performed three exercises with and without abdominal contraction: breathing, coughing and the Valsalva maneuver, with an average of 157, 107 and 108 images per subject and per exercise, respectively. Approximately 8% of the total number of slices were manually segmented by a single expert using FSLeyes software. The resulting segmented masks were then propagated to the remaining slices using an automatic label propagation algorithm based on image registration (Ogier *et al.*, 2017). This set of 7,293 axial images was segmented with four distinct labels corresponding to left and right rectus abdominis (RA) and lateral muscles (LM, which include transverse abdominis, internal and external obliques), as shown Figure 1-b. This segmentation technique has already been validated (Jourdan *et al.*, 2021), and the resulting masks serve as the ground truth for evaluating the quality of the segmentation performed by the nnUNet network.

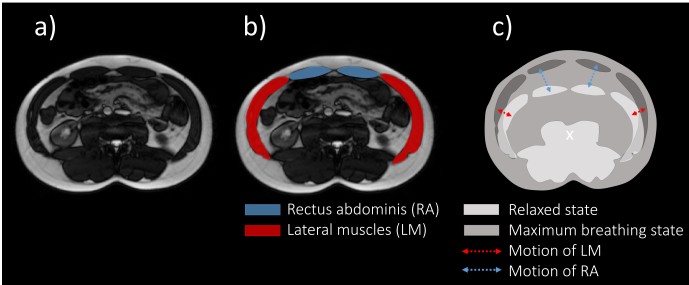

Figure 1: 2D axial abdominal MRI a) Without annotation b) Segmentation mask of the rectus abdominis (RA) and lateral muscles (LM) c) Motion quantification of the muscles

### 2.2. Neural network

nnUNet[1] (V2) for deep learning segmentation with automatic learning hyperparameter selection was used. Parameters such as batch size and network depth were determined

---

1. https://github.com/MIC-DKFZ/nnUNet

based on GPU size and input image size. Additional parameters were selected through 5-fold cross-validation (Isensee *et al.*, 2021). Traditional data augmentation methods were applied, excluding horizontal flip as this caused issues due to the symmetrical nature of the abdominal muscles. The data was decomposed into 416×416 frames, resampled to 256×256, and randomly split into a 15-subject training set (5,492 images) and a 5-subject test set (1,801 images). Training was conducted with PyTorch 1.2 on an NVIDIA GeForce RTX 2080 Ti GPU, with a training time of approximately 55 hours per fold.

To evaluate nnUNet's ability to segment abdominal wall muscles, DICE coefficient and Haussdorf distance were calculated between the ground truth masks and the nnUNet segmented masks. Additionally, the motion of the muscles (Jourdan *et al.*, 2021), as shown in Figure 1-c, from both the ground truth and nnUNet segmentation masks were compared.

## 3. Results

Figure 2 shows the DICE coefficient and Haussdorf distance of each label. The nnUNet performs better on the LM muscles than the RA muscles, likely due to the increased motion of the RA muscles during the exercises. The DICE is quite similar throughout the different exercises. The Haussdorf distance is higher for LM, this muscle being bigger than RA. The difference of motion between the ground truth and nnUNet masks is lower than 0.8 mm for both LM and RA, for the three exercises.

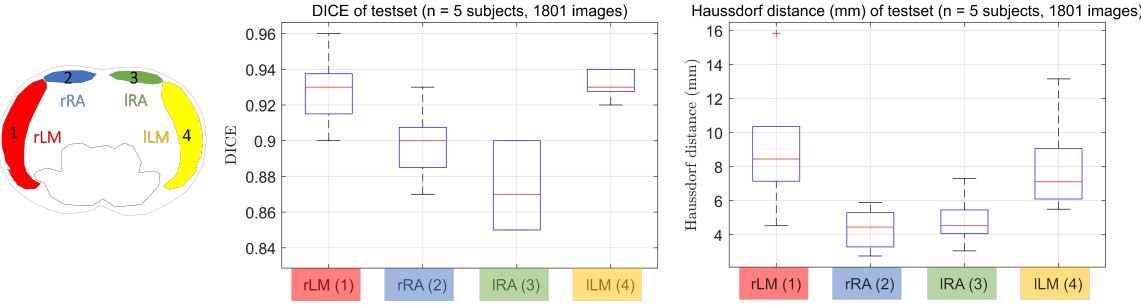

Figure 2: DICE and Haussdorf distance results depending on the label

## 4. Discussion and conclusion

In conclusion, this study demonstrates the effectiveness of nnUNet in segmenting dynamic axial MRI images of healthy subjects, which could significantly reduces researchers' manual workload and accelerates research progress. Moving forward, enhancing the network's performance will involve diversifying the training set with additional axial planes, thereby reducing dependence on body location. Increasing the number of subjects will add anatomical variability, while incorporating more exercises will capture greater dynamics and motion, ultimately enhancing the network's robustness and applicability. It's worth noting that this study represents ongoing work, with the ultimate goal of applying these advancements to segment patients' images. The MRI dataset presented in this study will be made available upon request on a public repository.

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
