# OpenReview forum: "Automatic muscle segmentation on healthy abdominal MRI using nnUNet"
_MIDL.io/2024/Short_Papers — MIDL 2024 Short Papers_

### Official Review · Reviewer_43Re · 2024-04-24

**Confidence:** 5
**Final Rating:** 4

**Review:**

The author trained a nnUnet method to segment 4 abdominal muscles in a inhouse MRI dataset. The application is new in MRI and the evaluation is sufficient. I would suggest to provide more information about the annotation process, e.g., who annotated the data? and how ground truth was verified? It would be better if the author can release the dataset.

---

### Decision · Program_Chairs · 2024-04-26

Accept